# Increased vs. Standard Dose of Iron in Ready-to-Use Therapeutic Foods for the Treatment of Severe Acute Malnutrition in a Community Setting: A Systematic Review and Meta-Analysis

**DOI:** 10.3390/nu14153116

**Published:** 2022-07-29

**Authors:** Aamer Imdad, Jaimie L. Rogner, Melissa François, Shehzad Ahmed, Abigail Smith, Olivia J. Tsistinas, Emily Tanner-Smith, Jai K. Das, Fanny F. Chen, Zulfiqar Ahmed Bhutta

**Affiliations:** 1Division of Pediatric Gastroenterology, Hepatology and Nutrition, Department of Pediatrics, SUNY Upstate Medical University, Syracuse, NY 13210, USA; 2Departments of Medicine and Pediatrics, University of Rochester Medical Center, Rochester, NY 13210, USA; jaimielrogner@gmail.com; 3College of Medicine, SUNY Upstate Medical University, Syracuse, NY 13210, USA; francome@upstate.edu (M.F.); chenf@upstate.edu (F.F.C.); 4Department of Pediatrics, SUNY Upstate Medical University, Syracuse, NY 13210, USA; ahmedshe@upstate.edu; 5Health Science Library, SUNY Upstate Medical University, Syracuse, NY 13210, USA; smithab@upstate.edu (A.S.); tsistijo@upstate.edu (O.J.T.); 6Department of Counseling Psychology and Human Services, College of Education, University of Oregon, Eugene, OR 97403, USA; etanners@uoregon.edu; 7Department of Pediatrics and Child Health and Institute of Global Health and Development, Aga Khan University, Karachi 74800, Pakistan; jai.das@aku.edu; 8Department of Global Child Health, Hospital for SickKids, Toronto, ON M5G 0A4, Canada; zulfiqar.bhutta@aku.edu; 9Center for Excellence in Women and Child Health, Aga Khan University, Karachi 74800, Pakistan

**Keywords:** severe acute malnutrition (SAM), ready-to-use therapeutic foods (RUTF), iron, anemia

## Abstract

The optimal dose of iron in ready-to-use therapeutic foods (RUTF) used to treat uncomplicated severe acute malnutrition (SAM) in community settings is not well established. The objective of this systematic review was to assess if an increased iron dose in RUTF, compared with the standard iron dose in the World Health Organization (WHO)-recommended peanut-based RUTF, improved outcomes in children aged six months or older. We searched multiple electronic databases and only included randomized controlled trials. We pooled the data in a meta-analysis to obtain relative risk (RR) and reported it with a 95% confidence interval (CI). Three studies, one each from Zambia, the Democratic Republic of Congo, and Malawi, were included. In all studies, the RUTF used in the intervention group was milk-free soya–maize–sorghum-based RUTF. The pooled results showed that, compared to the control group, a high iron content in RUTF may lead to increase in hemoglobin concentration (mean difference 0.33 g/dL, 95% CI: 0.02, 0.64, two studies, certainty of evidence: low) and a decrease in any anemia (RR 0.66, 95% CI: 0.48, 0.91, two studies, certainty of evidence: low), but also decrease recovery rates (RR 0.91, 95% CI: 0.84, 0.99, three studies, certainty of evidence: low) and increase mortality (RR 1.30, 95% CI: 0.87, 1.95, three studies, certainty of evidence: moderate). However, the CIs were imprecise for the latter outcome. Future studies with large sample sizes are needed to confirm the beneficial versus harmful effects of high iron content in RUTF in treating uncomplicated SAM in children aged 6-59 months in community settings.

## 1. Introduction

The World Health Organization (WHO) reports that about 45 million children under the age of five worldwide suffered from wasting (low weight for height) in the year 2020 [1]. Children with severe acute malnutrition (SAM) have a 5–20-fold higher risk of mortality than well-nourished children and are at a greater risk for immunodeficiency and neurodevelopmental delay [2]. Children with SAM have several key macronutrient and micronutrient deficiencies [2,3]. Previous studies have shown higher rates of anemia in children suffering from severe malnutrition with prevalence rates of 40% to 90%, and about half of these cases of anemia were attributed to iron deficiency [3,4,5,6]. Iron deficiency in severely malnourished children could be due to low intake, increased losses, higher demand, or poor absorption. Iron is essential for adequate catch-up growth and neurological development in children with SAM. Iron deficiency anemia may cause symptoms such as severe fatigue and it is a risk factor for severe complications such as heart failure and death. Iron deficiency anemia can also have long-term effects such as neurodevelopmental delay, especially in children less than two years of age [4,7,8].

The current standard of care for children with uncomplicated SAM involves using ready-to-use therapeutic food (RUTF) to promote growth [2,9]; however, the precise formulation to achieve optimal recovery remains unclear [10]. Moreover, recent studies suggest alternative RUTF formulations with a high iron content may be more effective in correcting anemia and iron deficiency in children with uncomplicated SAM compared to the current standard RUTF formulations [4,11,12]. In addition to the need for sufficient iron intake to treat iron deficiency, it remains unclear whether improving iron status among undernourished children increases the risk of complications, including the risk of morbidities such as diarrhea and malaria, or undesired changes in the microbiome [13,14,15]. This aspect is especially of concern in malaria-endemic regions in Africa because nearly one-third (27%) of all children affected by wasting worldwide reside in Africa [15,16]. Due to the critical role of RUTF in the treatment of children with uncomplicated SAM, more information is needed to generate formulations with optimal iron levels for the treatment of anemia and optimal growth and development in children suffering from malnutrition [7,8,11]. Data from randomized controlled trials have been recently available to address the optimal dose of iron in RUTF. Therefore, this review aimed to synthesize the most recent research on the iron content in RUTF in treating uncomplicated SAM in children aged 6–59 months in the community setting.

## 2. Materials and Methods

This systematic review was commissioned by the WHO, conducted according to the guidelines of the Cochrane Collaboration, and reported according to PRISMA 2020 guidelines [17]. The detailed protocol is published [18]. We describe the methods briefly in the sections below.

### 2.1. Inclusion/Exclusion Criteria

We considered only randomized trials that examined children aged six months or older with SAM being managed in outpatient settings, irrespective of HIV status. We included studies that defined SAM based on weight-for-height Z-scores (<−3 SD for WHO growth standards), mid–upper arm circumference (<115 mm), or the presence of bilateral edema [19]. The intervention of interest was the use of RUTF with a high iron content (iron content > the WHO standard iron content of 1.9 mg/100 kcal [10–14 mg/100 g] of RUTF) for the treatment of SAM in community settings. We included studies irrespective of the type of RUTF used, i.e., standard peanut based RUTF vs. low-milk-based vs. non-milk-based vs. locally prepared. The comparison group consisted of children receiving RUTF containing iron at the current WHO recommended level of 1.9 mg/100 kcal (10–14 mg/100 g) to treat SAM.

### 2.2. Outcomes

The primary outcomes of interest were blood hemoglobin concentration (g/dL), any anemia, severe anemia, iron deficiency anemia, recovery from SAM, and any adverse events. The secondary outcomes were all-cause mortality, clinical deterioration necessitating referral to inpatient care, withdrawal from the trial, relapse, serum ferritin level, serum zinc level, serum copper level, serum iron level, weight for age (kg or Z-scores), height for age (cm or Z-scores), weight-for-height Z-scores, microbiome outcomes of alpha diversity and beta diversity, and neurodevelopmental outcomes.

### 2.3. Literature Search

We conducted systematic electronic queries using key terms in several databases, including PubMed, EMBASE, the Cochrane Central Register for Controlled Trials, Web of Science, CINHAL, Scopus, LILACS, and the WHO Global Index Medicus. The last date of the search was 24 June 2021, and the search strategies used for various electronic databases are available in Appendix B. We did not apply any search restrictions to exclude studies based on the outcome, publication year, publication status, or language. Additional resources were also searched, as described in the published protocol [18]. The reference sections of the included studies and known systematic reviews on the topic were also hand-searched to include any eligible studies [9].

### 2.4. Data Extraction and Synthesis

The titles and abstracts of all the available studies from the literature searches were screened in duplicate with the help of Covidence14 software [20]. Two authors independently extracted the following information from the data: study design, study site (country/region), study year, study type, intervention, exposure, comparison, outcomes, and risk of bias. Studies with multiple treatment arms were included, if eligible. For multiple-arm trials, we included data such that the only difference between the groups was the dose of iron in the RUTF. The risk of bias in studies was evaluated using Version 2 of the Cochrane risk-of-bias tool for randomized trials (RoB 2.0) [21]. We conducted a random-effects meta-analysis when an outcome was reported by at least two studies, with the help of RevMan-5 software [22]. Dichotomous outcomes were assessed using relative risk effect sizes and presented with 95% confidence intervals (CIs). For continuous outcomes, we pooled the data to obtain an average mean difference and reported it with its 95% CI. Statistical heterogeneity of effect sizes within any given meta-analysis was assessed using the χ^2^, *I*^2^, and tau statistics [18].

The Grading of Recommendations, Assessment, Development and Evaluations (GRADE) approach was used to evaluate overall evidence quality using the software GRADEpro [23]. The GRADE approach is a comprehensive framework used to assess the overall quality of evidence for an outcome using characteristics such as study design, heterogeneity, directness of evidence, risk of bias, publication bias, and precision of effect estimates [24]. The results of the GRADE assessment were included in a GRADE evidence profile table.

### 2.5. Subgroup Analyses and Sensitivity Analyses

We planned a number of subgroup analyses [18]. based on country: low-income country vs. middle-income country vs. high-income country; type of RUTF: standard RUTF with high iron content vs. non-standard RUTF with high iron content; type of participants: studies that included children with HIV vs. studies that did not include children with HIV; age: <24 months vs. 24–59 months vs. >59 months; hospitalization: children hospitalized (due to medical complication) prior to starting RUTF vs. children not hospitalized prior to starting RUTF; iron compound (formulation/chemical compound and amount); dose: intervention groups with a dose of iron higher than the WHO standard RUTF vs. intervention group receiving a dose lower the WHO standard RUTF; anemia status: children with anemia at baseline vs. those without anemia at baseline; and time of follow-up: 1 month vs. 3 months vs. 6 months follow-up, and longest follow-up [18]. None of these subgroup analyses were performed because the number of included studies was small. We also planned sensitivity analyses based on studies with a high risk of bias and the type of model used: random vs. fixed-effect model. Only the latter was performed due to the limited number of studies.

### 2.6. Patient and Public Involvement

No patient or public involvement was considered in the preparation of the protocol or the review.

## 3. Results

### 3.1. Literature Search

The literature search identified 393 titles after the exclusion of duplicates. Figure 1 shows the results of the literature search. After screening the full texts of 19 studies for eligibility, we ultimately included three studies [4,12,25]. reported in five publications [4,12,25,26,27]. (complete list in Appendix A). We excluded 14 studies, and the reasons for exclusion can be found in Appendix A.

### 3.2. Characteristics of Included Studies

Table 1 and Table 2 display the characteristics of the included studies for participants and interventions, respectively.

All three included studies were randomized controlled trials conducted in Africa. The studies’ median sample size was 886 [12]. participants, ranging from 392 [4]. participants to 1927 [25]. participants. All studies investigated an intervention using RUTF with an iron content higher than that of the current WHO standard RUTF for treatment of SAM in children aged 6–59 months in community settings. In all studies, the RUTF with a high iron content was soya–maize–sorghum (SMS)-based. The macronutrient and micronutrient compositions of the intervention- and comparison-group RUTF for each study are described in Table 3.

### 3.3. Studies with Multiple Intervention Arms and Missing Data

One study had two treatment arms: soybean, maize and sorghum (SMS), with milk (MSMS-RUTF) and without milk (FSMS-RUTF) [4]. We used the data from the (FSMS-RUTF) group only, as the other two included groups also used an RUTF that was non-dairy SMS-based. One of the included studies was a cluster randomized trial [25]. The sample size of this cluster randomized trial was adjusted for the cluster design [25], so we did not make any further adjustments. Even though all the studies contributed data for the meta-analysis, they did not all contribute to every outcome in the review.

### 3.4. Effects of Interventions

In the section below, we report the meta-analysis and GRADE analysis results for each primary outcome and key secondary outcomes at the longest follow-up. Table 4 shows GRADE evidence profiles for the same outcomes.

### 3.5. Primary Outcomes

#### 3.5.1. Blood Hemoglobin Concentration (g/dL) at the Longest Follow-Up

Two randomized controlled trials [4,12]. reported data on blood hemoglobin concentration, and both had data available only for a subset of the population. Data for hemoglobin at the end of the individual studies were pooled from these two studies, which included 451 participants, with 219 participants in the high-iron RUTF group and 232 in the WHO standard-iron RUTF group. The results showed low certainty evidence that there may be a small increase in hemoglobin concentration among children aged 6–59 months receiving RUTF with a high iron content, compared to those receiving RUTF with the WHO standard iron dose (mean difference 0.33 g/dL, 95% CI: 0.02, 0.64, *p* = 0.04, *I*^2^ = 52%, Figure 2). We downgraded the GRADE certainty for risk of bias (because both of the included studies [4,12] reported data for only a subset of the study population, Appendix A) and imprecision of the summary estimate (the overall effect seems to be small; the lower limit of the confidence interval was very close to the null effect, and the results of the blood hemoglobin were not adjusted for ethnicity and altitude) (Table 4).

#### 3.5.2. Subgroup and Sensitivity Analyses

The a priori subgroup analyses planned for age group (<24 months vs. 24–59 months vs. >59 months), country, type of RUTF, type of participants (studies that included children with HIV vs. studies that did not include children with HIV), hospitalization, iron compound, anemia status, and time of follow-up were not conducted because there were not enough data in the included studies. Sensitivity analysis based on the type of model used showed similar results for the fixed vs. random effects model of the meta-analysis (MD: 0.31, 95% CI: 0.10, 0.52, fixed models).

#### 3.5.3. Any Anemia at the Longest Follow-Up

Two randomized controlled trials [4,12]. reported data on anemia, and both had data available only for a subset of the population. Anemia was defined in both studies [4,12]. as blood hemoglobin < 11 mg/dL. The data on hemoglobin from one study [12]. were not available from the published report but were provided upon request by the authors; these data were not adjusted for altitude and ethnicity. The published data from the second study [4]. were adjusted for altitude and ethnicity; however, we had access to unadjusted data. We pooled the same to be consistent with the data from the first study. Data for any anemia at the end of the study were pooled from these two studies, which included 451 participants, with 219 participants in the high-iron RUTF group and 232 in the WHO standard-iron RUTF group. The results showed a low certainty of evidence that the risk of anemia may be lower in children aged 6–59 months receiving RUTF with a high iron content, compared to those receiving RUTF with the WHO standard iron dose (RR 0.66, 95% CI: 0.48, 0.91, *p* = 0.01, *I*^2^ = 32%, Figure 3). We downgraded the GRADE certainty for risk of bias (for which there was ‘serious concern’ for a high risk of bias because both of the two included studies [4,12]. reported data for only a subset of the study population) and imprecision (for which there was ‘serious concern’ because the confidence interval around the summary estimate almost approached the null effect and the values of hemoglobin were not adjusted for altitude and ethnicity) (Table 4).

### 3.6. Subgroup and Sensitivity Analyses

None of the a priori subgroup analyses were performed for this outcome due to the lack of available data in the included studies. A sensitivity analysis based on the type of model showed similar results (RR 0.65, 95% CI: 0.50, 0.85. fixed effects).

#### 3.6.1. Iron Deficiency Anemia at the Longest Follow-Up

One randomized controlled trial [4]. reported data on iron deficiency anemia; data were only available for a subset of the population. The hemoglobin results were adjusted for altitude and ethnicity. Data for iron deficiency anemia at the end of the study included 146 participants, with 63 participants in the high-iron RUTF group and 83 in the WHO standard-iron RUTF group. The results showed a low certainty of evidence that the risk of iron deficiency anemia may be lower in children aged 6–59 months receiving RUTF with a high iron content compared to those receiving RUTF with the WHO standard iron dose (RR 0.39, 95% CI: 0.15, 0.99, *p* = 0.05, Figure 4). We downgraded the GRADE certainty for risk of bias (for which there was ‘serious concern’ for a high risk of bias because the study [4]. reported data for only a subset of the study population) and imprecision (for which there was ‘serious concern’ because the analysis included only one study with a total of 22 events in both groups, the confidence interval around the summary estimate was imprecise, and the upper limit of the confidence interval almost reached a null effect) (Table 4).

#### 3.6.2. Subgroup and Sensitivity Analyses

None of the a priori subgroup analyses or sensitivity analyses were performed for this outcome.

#### 3.6.3. Severe Anemia (Hemoglobin < 9 mg/dL) at the Longest Follow-Up

Two randomized controlled trials [4,12] reported data on severe anemia, and both had data available only for a subset of the population. Data for severe anemia at the end of the study were pooled from these two studies, which included 451 participants, with 219 participants in the high-iron RUTF group and 232 in the WHO standard-iron RUTF group. The results showed a low certainty of evidence that risk of severe anemia might be lower in children aged 6–59 months receiving RUTF with a high iron content, compared to those receiving RUTF with the WHO standard iron dose (RR 0.88, 95% CI: 0.30, 2.56, *p* = 0.81, *I*^2^ = 0%, Figure 5). We downgraded the GRADE certainty for risk of bias (for which there was ‘serious concern’ for high risk of bias because both of the two included studies [4,12]. reported data for only a subset of the study population) and imprecision (the number of events were small, and the confidence interval of the summary estimate included a null effect).

#### 3.6.4. Subgroup and Sensitivity Analyses

None of the a priori subgroup analyses were performed for this outcome and a sensitivity analysis-based model showed similar results (RR 0.88, 95% CI 0.30, 2.56. fixed effects).

#### 3.6.5. Recovery from SAM at the Longest Follow-Up

Three randomized controlled trials [4,12,25]. reported data on recovery from SAM. Raw values were used to calculate the summary estimate from individual studies, and an intention-to-treat analysis was preferred, where available. Data for recovery from SAM at the end of the study were pooled from these three studies, which included 3681 participants, with 1696 participants in the high-iron RUTF group and 1985 in the WHO standard-iron RUTF group. The results showed a low certainty of evidence that the rate of recovery may be lower for children aged 6–59 months receiving SMS-based RUTF with a high iron content, compared to those receiving RUTF with the WHO standard iron dose (RR 0.91, 95% CI: 0.84, 0.99, *p* = 0.04, *I*^2^ = 76%, Figure 6). We downgraded the GRADE certainty for inconsistency (for which there was ‘serious concern’ because even though the magnitude of effect differed among the included studies, with an *I*^2^ of 76%) and imprecision (for which there was ‘serious concern’ because the upper limit of the confidence interval around the summary estimate almost reached a null effect) (Table 4).

#### 3.6.6. Adverse Events

Three randomized controlled trials [4,12,25]. reported data on adverse effects. The data were reported so that it could not be meta-analyzed, so we present the results in Table 5. None of these studies reported outcomes regarding the impact of high iron content in RUTF on the incidence of diarrhea and malaria; thus, we could not explore these outcomes, as initially intended. There was no significant difference in the rates of any adverse events or serious adverse events between the SMS-based high-iron RUTF group vs. the control group in any of the included studies. One study [25]. reported that a skin rash occurred in 13.3% of children in the group receiving peanut-based RUTF with the WHO standard iron dose, compared to no skin rash occurring among children in the group receiving RUTF with a high iron content; however, it was noted that all of the children reporting a skin rash were from the same health center, and the rash was mild.

### 3.7. Secondary Outcomes

We describe the results of secondary outcomes, for which a GRADE analysis was conducted, in Table 4. The data for the following secondary outcomes were not available: serum zinc, serum copper, relapse, clinical deterioration necessitating referral to inpatient care, height for age, weight for height, microbiome outcomes, and neurodevelopmental outcomes.

#### 3.7.1. All-Cause Mortality at the Longest Follow-Up

Three randomized controlled trials [4,12,25]. reported data on all-cause mortality. Raw values from individual studies were used to calculate the relative risk, and an intention-to-treat analysis was preferred, where available. Data for all-cause mortality at the end of the study were pooled from these three studies, which included 3686 participants, with 1696 participants in the high-iron RUTF group and 1990 in the WHO standard-iron RUTF group. The results showed a moderate certainty of evidence that there may be an increase in all-cause mortality for children aged 6–59 months receiving SMS-based RUTF with a high iron content, compared to those receiving peanut-based RUTF with the WHO standard iron dose (RR 1.30, 95% CI: 0.87, 1.95, *p* = 0.21, *I*^2^ = 24%, Figure 7); however, a potential beneficial effect cannot be ruled out based on the lower limits of the confidence interval. We downgraded the GRADE certainty for imprecision (for which there was ‘serious concern’ because the confidence interval around the summary estimate included a null effect with the possibility of a beneficial effect or a decreased risk of mortality) (Table 4).

#### 3.7.2. Withdrawal from Trial

Three randomized controlled trials [4,12,25]. reported data on withdrawal from the study, which included 3681 participants, with 1696 participants in the SMS-based high-iron RUTF group and 1985 in the WHO standard-iron peanut-based RUTF group. The results showed a low certainty of evidence that the rates of withdrawal may be higher in children aged 6–59 months receiving SMS-based RUTF with a high iron content, compared to those receiving peanut-based RUTF with the WHO standard iron dose (risk ratio 1.25, 95% CI: 0.98, 1.60, *p* = 0.08, *I*^2^ = 60%, Figure 8). We downgraded the GRADE certainty for inconsistency (for which there was ‘serious concern’ because the magnitude of the effect varied among the studies, with an *I*^2^ of 60%) and imprecision (for which there was ‘serious concern’ because the confidence interval around the summary estimate included a null effect with the possibility of a beneficial effect or a decreased risk of withdrawal from the trial) (Table 4).

#### 3.7.3. Weight Gain

Two randomized controlled trials [4,12]. reported data on weight gain and the pooled data from these two studies showed that the rate of weight gain was lower among children aged 6–59 months receiving RUTF with a high iron content, compared to those receiving RUTF with the WHO standard iron dose (mean difference −0.56 g/dL, 95% CI: −01.61, −0.49, *p* = 0.003, *I*^2^ = 88%, Figure 9).

## 4. Discussion

### 4.1. Summary of Main Results

This systematic review and meta-analysis evaluated the effect of a high iron content in SMS-based RUTF versus the iron content in peanut-based RUTF based on current WHO standard guidelines in children aged 6-59 months with uncomplicated SAM. Results from the synthesis suggest low certainty evidence that the use of SMS-based high-iron RUTF, compared to the WHO standard iron content peanut-based RUTF, may lead to an increased blood hemoglobin concentration and decreased risk of any anemia, iron deficiency anemia, and severe anemia. However, low certainty evidence showed that recovery rates may be lower in the high-iron SMS-based RUTF group compared to standard-iron peanut-based RUTF in children aged 6–59 months with SAM. A moderate level of certainty of evidence showed that the mortality risk may be higher in the high-iron group, although the confidence interval also included a potentially beneficial effect. The available data on rates of side-effects showed similar rates in the intervention group compared to the control.

### 4.2. Overall Completeness of Evidence

This review summarized evidence from three RCTs comprising 3205 participants; however, data were not available from all included studies for all outcomes considered in this review. All three studies reported data for mortality and recovery rates, but the data on hemoglobin-related outcomes were available from only two studies and only for a subset of the populations in these studies. Additionally, no data were available regarding the safety outcomes of interest in this review, i.e., clinical deterioration requiring hospitalization. There were also not enough studies to perform all the a priori subgroup analyses, so no conclusions can be drawn at this time for any differential effects of a high iron content in RUTF based on age group (<24 months vs. 24–59 months vs. >59 months), country income level, type of RUTF, type of participants (studies that included children with HIV vs. studies that did not include children with HIV), hospitalization, iron compound, anemia status, or duration of follow-up. Furthermore, each of the three included studies utilized a different dose of iron in the interventional RUTF; therefore, there were not enough data available to perform any statistical analyses to elucidate a dose–response relationship and determine an optimal iron dose in RUTF.

### 4.3. Certainty of Evidence

The certainty of the evidence was graded as low for all the primary outcomes and most of the secondary outcomes (Table 4). The most common reasons for downgrading the evidence for the primary outcomes were high risk of bias and imprecision. The data were available from only a subset of the study population, and the number of events were small for most of the outcomes. Even though all the studies were conducted in Africa, we did not downgrade for indirectness, as all the included studies had children aged 6–59 months with SAM being managed in community settings. However, we think that the results of the available evidence should be replicated in additional sites in Africa and in Southeast Asia, where the burden of SAM is very high. The pooled data were homogenous for most of the outcomes except the outcomes of recovery from SAM and withdrawal from the studies, in which significant statistical heterogeneity was noted, and the certainty of the evidence was downgraded accordingly.

### 4.4. Potential Bias in the Review Process

We followed the standardized methods of the Cochrane Collaboration to conduct this review. We wrote a protocol for the review that was externally reviewed and publicly available [18]. All titles and abstracts were screened in duplicate, and data extraction was also performed in duplicate for the included studies. We used Version 2 of the Cochrane risk-of-bias tool for randomized trials (RoB 2), a newly developed tool to assess the risk of bias for each outcome from a study rather than a risk of bias assessment applied to all the outcomes from that study (Appendix A). This approach allowed us to give one risk of bias assessment for certain hemoglobin-related outcomes, for which data were available from only a subset of study participants for two of the included studies (high), and a different risk of bias assessment (low) for other outcomes such as mortality and recovery rates, for which data were available from all the study participants from these studies. The authors provided data on some of the hemoglobin-related outcomes; however, these data were not adjusted for altitude and ethnicity. The data from the other study were adjusted; however, the unadjusted data were available from the primary authors. Thus, we decided to pool the unadjusted data to be consistent and adjusted the certainty of evidence in the GRADE analysis. One of the included studies had two intervention groups that used a high iron content. We used the data from one of the groups, only because the other two included studies had a similar composition of the RUTF—non-dairy, non-peanut-based, and based on soya–maize–sorghum (SMS-based)—which was locally available for the study populations. To the best of our knowledge, there has been no other systematic review published on this topic so we could not compare our findings with other published reviews.

### 4.5. Implications for Practice

The low certainty evidence synthesized in this systematic review showed rates of anemia may be lower and hemoglobin levels may be higher in the intervention group that consumed SMS-based RUTF with a high iron content, compared to the group that consumed peanut-based RUTF with standard WHO-recommended iron content, in children aged 6–59 months with SAM. However, the recovery rates from SAM in the SMS-based high-iron RUTF group were lower, and there was a potentially higher risk of mortality in this group compared to the standard peanut-based RUTF. The lower recovery rates seen in the SMS-based high-iron RUTF group were consistent with those demonstrated in a previous review, which found all non-dairy RUTFs [including soya–maize–sorghum (SMS)-based RUTFs]. were associated with lower recovery rates than standard peanut-based RUTF in children with SAM [28]. Thus, the lower recovery rates noted in the included studies in this review could be due to SMS-based macronutrients rather than the iron content of the RUTF. All the included studies used SMS-based RUTF for the high-iron RUTF, so the results cannot be generalized to other forms of RUTF such as those that are peanut-based. Further studies will be required before a solid conclusion can be drawn regarding increasing the iron content in RUTF for the treatment of uncomplicated SAM in children aged 6–59 months.

### 4.6. Implications for Research

Two of the three included studies reported data on blood hemoglobin-related outcomes for a subset of participants. To support the WHO’s aim to establish the optimal dose of iron in RUTF for the treatment of children with uncomplicated SAM, there is a need for large randomized controlled trials, preferably utilizing similar RUTF recipes, in which the only difference between formulations is the dose of iron. The studies should be designed to demonstrate a dose–response relationship as well as to assess safety, such as by measuring the number of adverse events and incidence of malaria and diarrhea. The possible increased risk of mortality in the high-iron SMS-based RUFT group also warrants further investigation.

## 5. Conclusions

The use of SMS-based RUTF with a high iron content for treating uncomplicated SAM in children aged 6–59 months in community settings may lead to higher blood hemoglobin levels and lower rates of anemia and severe anemia than the use of RUTF with standard WHO-recommended iron content; however, the certainty of the evidence is low for these findings. There is a potential increase in mortality and a decrease in recovery rates in children provided with an SMS-based high-iron RUTF compared to those provided with a peanut-based RUTF with the WHO standard iron dose. Future studies with a large sample size are needed to confirm the beneficial versus harmful effects of a high iron content in RUTF in treating uncomplicated SAM in children aged 6–59 months in community settings.

## Figures and Tables

**Figure 1 nutrients-14-03116-f001:**
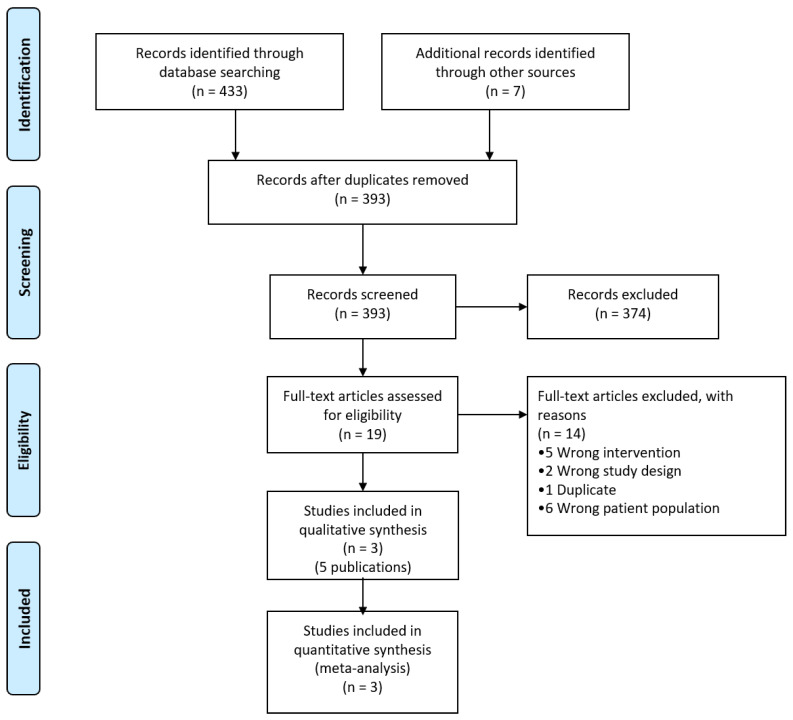
PRISMA diagram outlining the literature search. All searches were carried out on 24 June 2021.

**Figure 2 nutrients-14-03116-f002:**
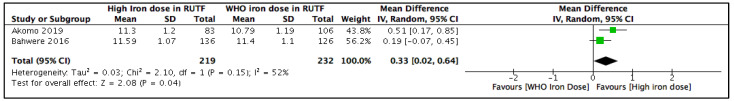
Effect of SMS-based RUTF with high iron content compared to that of peanut-based RUTF with WHO standard iron content on blood hemoglobin (g/dL). The forest plot shows pooled data from two studies for change in hemoglobin at the end of the study. Both the studies had data available only for a subset of population. The published data from Akomo 2019 [4]. was adjusted for altitude and ethnicity; however, we had access to unadjusted data, and we pooled the same to be consistent with data from Bahwere’s 2016 [12]. study that were not adjusted. The data from Bahwere 2016 was provided by authors and was not available from the published report. Akomo 2019 has two study groups. We included data from milk-free soybean, maize, and sorghum (FSMS) and a standard formulation prepared from peanuts and milk (PM-RUTF). Abbreviations: RUTF—ready-to-use therapeutic food; WHO—World Health Organization; SMS—soya–maize–sorghum.

**Figure 3 nutrients-14-03116-f003:**
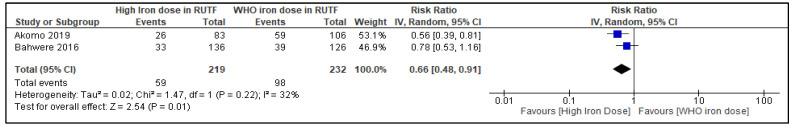
Effect of SMS-based RUTF with high iron content compared to peanut-based RUTF with WHO standard iron content on any anemia. The forest plot shows pooled data from two studies. The data on hemoglobin from Bahwere 2016 [12]. was not adjusted for altitude and ethnicity. The published data from Akomo 2019 [4]. was adjusted for altitude and ethnicity; however, we had access to unadjusted data and we pooled the same to be consistent with data from Bahwere’s 2016 study. The data from Bahwere 2016 was provided by authors and was not available from the published report. Akomo 2019 has two study groups. We included data from milk-free soybean, maize and sorghum (FSMS) and a standard formulation prepared from peanuts and milk (PM-RUTF). The overall results indicate that risk of anemia might be lower in group that received RUTF with high iron content. Abbreviations: RUTF—ready-to-use therapeutic food; WHO—World Health Organization; SMS—soya–maize–sorghum.

**Figure 4 nutrients-14-03116-f004:**
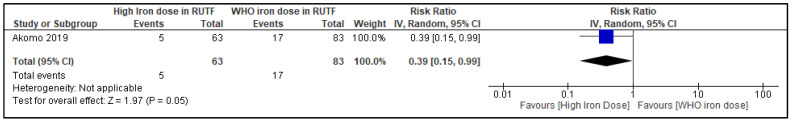
Effect of SMS-based RUTF with high iron content compared to peanut-based RUTF with WHO standard iron content on iron deficiency anemia. The forest plot shows data from a single study and this study had data available only for a subset of population. Akomo 2019 [4]. has two study groups. We included data from milk-free soybean, maize and sorghum (FSMS) and a standard formulation prepared from peanuts and milk (PM-RUTF). Abbreviations: RUTF—ready-to-use therapeutic food; WHO—World Health Organization; SMS—soya–maize–sorghum.

**Figure 5 nutrients-14-03116-f005:**
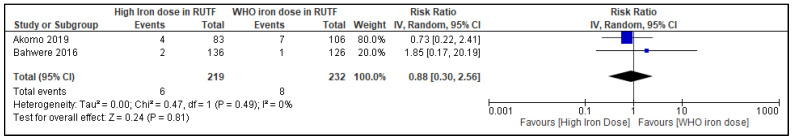
Effect of SMS-based RUTF with high iron content compared to peanut-based RUTF with WHO standard iron content on severe anemia (Blood Hemoglobin < 9 mg/dL). The forest plot shows pooled data from two studies. Data were available only for a subset of population from both the studies. The blood hemoglobin results were not adjusted for altitude and ethnicity. The data was provided by authors of the primary studies and was not available from the published papers. Akomo 2019 has two study groups. We included data from milk-free soybean, maize and sorghum (FSMS) and standard formulation prepared from peanut and milk (PM-RUTF). Abbreviations: RUTF—ready-to-use therapeutic food; WHO—World Health Organization; SMS—soya–maize–sorghum.

**Figure 6 nutrients-14-03116-f006:**
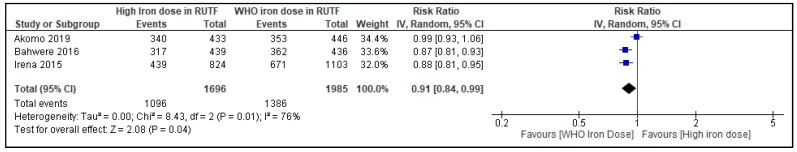
Effect of SMS-based RUTF with high iron content compared to peanut-based RUTF with WHO standard iron content on recovery from severe acute malnutrition. The forest plot shows pooled data from all three included studies from this review. Raw values were used, and an intention-to-treat analysis was preferred, where available. The pooled results show a potential decrease in recovery rates for children receiving RUTF with high iron content. Akomo 2019 [4]. has two study groups. We included data from milk-free soybean, maize and sorghum (FSMS) and a standard formulation prepared from peanuts and milk (PM-RUTF). Abbreviations: RUTF—ready-to-use therapeutic food; WHO—World Health Organization; SMS—soya–maize–sorghum.

**Figure 7 nutrients-14-03116-f007:**
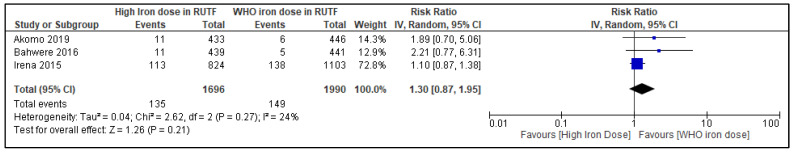
Effect of SMS-based RUTF with high iron content compared to that of peanut-based RUTF with WHO standard iron content on all-cause mortality. The forest plot shows pooled data from all three included studies from this review. Raw values were used, and an intention-to-treat analysis was preferred, where available. The pooled results show a potential increase in mortality for children receiving RUTF with high iron content, compared to RUTF with WHO standard iron content. Akomo 2019 [4]. has two study groups. We included data from milk-free soybean, maize and sorghum (FSMS) and a standard formulation prepared from peanuts and milk (PM-RUTF). Abbreviations: RUTF—ready-to-use therapeutic food; WHO—World Health Organization; SMS—soya–maize–sorghum.

**Figure 8 nutrients-14-03116-f008:**
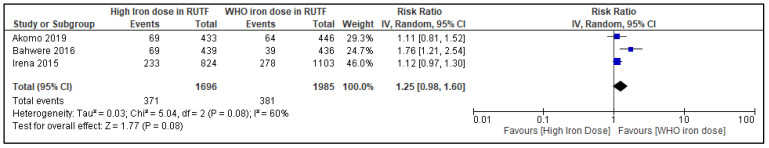
Effect of SMS-based RUTF with high iron content compared to that of peanut-based RUTF with WHO standard iron content on default from trial. The forest plot shows pooled data from all three included studies from this review. Raw values were used, and an intention-to-treat analysis was preferred, where available. The pooled results show a potential increase in default rate for children receiving RUTF with high iron content compared to RUTF with WHO standard iron content. Akomo 2019 [4]. has two study groups. We included data from milk-free soybean, maize and sorghum (FSMS) and a standard formulation prepared from peanuts and milk (PM-RUTF). Abbreviations: RUTF—ready-to-use therapeutic food; WHO—World Health Organization; SMS—soya–maize–sorghum.

**Figure 9 nutrients-14-03116-f009:**
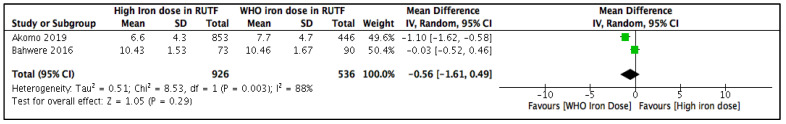
Effect of SMS-based RUTF with high iron content compared to that of peanut-based RUTF with WHO standard iron content on weight gain. The forest plot shows pooled data from two of the included studies. Raw values were used, and an intention-to-treat analysis was preferred, where available. The pooled results show a potential decrease in weight gain with RUTF with high iron content compared to RUTF with WHO standard iron content. Akomo 2019 [4]. has two study groups. We included data from milk-free soybean, maize and sorghum (FSMS) and a standard formulation prepared from peanuts and milk (PM-RUTF). Abbreviations: RUTF—ready-to-use therapeutic food; WHO—World Health Organization; SMS—soya–maize–sorghum.

**Table 1 nutrients-14-03116-t001:** Participant characteristics in the included studies.

Author	Type of Study	Country	Years of Data Collection	Total Number Randomized in All Study Groups	% Female	Inclusion Criteria/Exclusion Criteria ^a^	Notes
Irena 2015 [25]	RCT	Zambia	2009–2010	1927	47.9	Inclusion: SAM (MUAC < 11.0 cm or pitting edema) without complications. Exclusion: Previously included in the study	About 14.5% of the study population were HIV positive, 65% had edema
Bahwere 2016 [12]	RCT	Democratic Republic of Congo	2013–2014	886	47.1	Inclusion: SAM (MUAC < 115 mm or bilateral pitting edema assessed), Presence of appetite and no medical complications. Exclusion: Congenital or acquired disorder, food allergies, visiting families	About 20% had edema at baseline. Study site was highland plains and hills at elevations ranging between 900 and 1900 m.
Akomo 2019 [4]	RCT	Malawi	2015–2016	392	48.5	Inclusion: SAM (MUAC < 115 mm or bilateral pitting edema), good appetite and no medical complications. Exclusion: Parent refusal, congenital or acquired disorder, food allergies, visting families	About 54% had edema, 48% had anemia, and 33% had iron deficiency anemia at baseline. The altitude of the study area ranges from 578 to 1300 m above sea level.

^a^ all children were aged 6–59 months in the included studies. Abbreviations: HIV: human immunodeficiency virus, RCT: randomized controlled trial, SAM: severe acute malnutrition, MUAC: mid–upper arm circumference

**Table 2 nutrients-14-03116-t002:** Treatment characteristics in the included studies.

Author	Iron Dose ^a^ in the Experimental Group	Zinc Dose ^a^ in the Experimental Group	Frequency of RUTF	Duration of Intervention RUTF	Iron Content in Comparison Standard RUTF	Reported Outcomes
Irena 2015 [25]	SMS-RUTF 52.5	18.5	“1-week ration of RUTF and health and nutrition advice. Total calories 200 kcal/kg/day.”	2 weeks	12 mg/100 g RUTF	Recovery rates, mortality, default, non-recovered, mean rate of weight gain (g/kg/day), anemia
Bahwere 2016 [12]	SMS-RUTF 43.8	18.5	Ad libitum	–	11.1 mg/100 g RUTF	Recovery rates, mortality, mean daily weight gain, hemoglobin changes, plasma concentrations of eight key amino acids
Akomo 2019 [4]	FSMS-RUTF: 35.1 MSMS-RUTF: 31.6	FSMS-RUTF: 19.5 MSMS-RUTF: 19.9	Ad libitum	–	10.5 mg/100 g RUTF	Recovery rates, mortality, anemia

^a^ dose: mg/100 g of RUTF. Abbreviations: FSMS-RUTF: dairy-free soya–maize–sorghum ready-to-use therapeutic food, RUTF: ready-to-use therapeutic food. SMS-RUTF: soya–maize–sorghum-based ready-to-use therapeutic food. SAM: severe acute malnutrition

**Table 3 nutrients-14-03116-t003:** Macronutrient and micronutrient composition of RUTF in the included studies.

Ingredients	Unit ^	WHO Standard Peanut-Based RUTF	Irena 2015	Bahwere 2016	Akomo 2019	
			SMS-RUTF	SMS-RUTF	FSMS-RUTF	PM-RUTF
Soybean	g	0.0	29.7	38.6	–	–
Maize	g	0.0	18.2	4.0	–	–
Sorghum	g	0.0	6.5	10.0	–	–
Dried skim milk	g	25.0	0.0	0.0	–	–
Water	g	–	–	–	2.2	1.1
Ash	g	–	–	–	3.9	3.9
Sugar	g	27.4	14.6	16.7	22.5	25.0
Peanut paste	g	26.0	0.0	0.0	–	–
Palm oil	g	0.0	22.4	21.6	–	–
Soybean oil	g	20.0	0.0	–	–	–
Linseed oil	g	–	–	2.1	–	–
Palm stearin	g	0.0	5.6	4.0	–	–
Vitamin and minerals premix	g	1.6	3.0	3.0	2.5	1.6
*Nutrients*						
Energy	Kcal	530	521	553	532	545
Protein	g	15.9	11.1	16.5	18.4	15.6
Fat	g	33.0	33.0	36.3	34.2	33.8
Carbohydrate	g	–	55.0	–	41.3	45.0
Fibre	g	–	–	–	7.1	1.9
Protein/energy ratio	%	12	8.5	11.9	13.8	11.4
Fat/energy ratio	%	56.0	57.0	59.1	57.9	55.8
Omega-6/energy ratio	%	–	10.4	12.3	5.15	5.01
Omega-3/energy ratio	%	–	1.1	3.1	0.43	0.50
Omega-6/omega-3 ratio	%	–	9.6	4.0	12.0	10.0
SFAs	g	–	–	–	13.5	11.0
MUFAs	g	–	–	–	11.1	18.2
PUFAs	g	–	–	–	5.58	3.16
Trans fat	g	–	–	–	0.16	–
Vitamin A	µg	910	1852	1000	–	–
	mg RE	–	–	–	1.25	1.18
Vitamin C	mg	53	139	329	323	87
Vitamin D	µg	16	14	14	19.2	18.7
Vitamin E	mg	20	139	40.7	39	35
Thiamin (Vitamin B1)	mg	0.6	1.4	1.4	1.28	0.97
Riboflavin (Vitamin B2)	mg	1.8	1.9	1.9	1.63	3.20
Niacin (Vitamin B3)	mg	5.3	19	19	7.54	7.6
Pantothenic acid (Vitamin B5)	mg	3.1	8.3	8.3	5.36	4.5
Pyridoxine (Vitamin B6)	mg	0.6	1.4	1.4	0.99	0.66
Biotin (Vitamin B7)	µg	65	56	56	86	80
Folates (Vitamin B9)	µg	210	370	370	210	268
Cobalamin (Vitamin B12)	µg	1.8	2.3	4.3	2.5	3.2
Vitamin K	µg	21	14	14	26	22
Choline	mg	–	–	–	90	–
Calcium	mg	315	463	437.8	571	434
Phosphorus	mg	370	380	446.0	503	351
Magnesium	mg	86	74	74	104	97
Sodium	mg	–	–	–	87.3	131.4
Potassium	mg	1140	704	1155.8	991	1125
Copper	mg	1.7	0.9	0.9	1.48	1.60
Iodine	µg	100	417	417	100	85
Iron	mg	12	52.5	43.8	35.1	10.5
Zinc	mg	11.1	18.5	18.5	19.5	11.1
Selenium	µg	–	–	–	26	27
Manganese	mg	–	–	–	1.71	–
Phytic acid	mg	255	475	420	465	251
Phytic acid/zinc ratio	–	2.2	2.5	2.0	2.36	2.24
Phytic acid/iron ratio	–	1.9	0.8	0.8	1.12	2.02
Ascorbic acid/iron molar ratio	–	1.4	0.8	2.4	2.93	2.64
Ascorbic acid/iron weight ratio	–	4.4	2.6	7.5	9.20	8.29
Calcium/phosphorus weight ratio	–	0.9	1.2	1.0	1.14	1.24
Zinc/copper weight ratio	–	6.5	20.6	20.6	13.18	6.94
Zinc/iron weight ratio	–	0.9	0.4	0.4	0.56	1.06

Abbreviations: FSMS-RUTF: milk-free soya–maize–sorghum-based ready-to-use therapeutic food; MUFA: monounsaturated fatty acid; P-RUTF/PM-RUTF: peanut-paste-based ready-to-use therapeutic food; PUFA: polyunsaturated fatty acid; RE: retinol equivalent; RUTF: ready-to-use therapeutic food; FA: saturated fatty acid; SMS-RUTF: soya–maize–sorghum-based ready-to-use therapeutic food; ^ values are/100 g unless specified otherwise.

**Table 4 nutrients-14-03116-t004:** GRADE evidence profile to show the certainty of evidence for the primary outcomes and selected secondary outcomes. ***Population:*** Children aged 6–59 months with SAM. ***Intervention:*** SMS-based RUTF with a high iron content. ***Comparison:*** Peanut-based WHO standard RUTF. ***Settings:*** Outpatient.

Certainty Assessment							No. of Patients		Effect		Certainty
No. of Studies	Study Design	Risk of Bias	Inconsistency	Indirectness	Imprecision	Other Considerations	High Iron Content	WHO Standard Iron Content	Relative Risk (95% CI)	Absolute (95% CI)	
**Blood Hemoglobin (mg/dL)**											
2	RCTs	serious ^a^	not serious ^b^	not serious	serious ^c^	none	219	232	-	MD **0.33 mg/dL higher** (0.02 higher to 0.64 higher)	⨁⨁◯◯ Low
**Any Anemia (Blood Hemoglobin < 11 mg/dL)**											
2	RCTs	serious ^d^	not serious	not serious	serious ^e^	none	59/219 (26.9%)	98/232 (42.2%)	RR 0.66 (0.48 to 0.91)	**144 fewer per 1000** (from 220 fewer to 38 fewer)	⨁⨁◯◯ Low
**Iron Deficiency Anemia**											
1	RCT	serious ^f^	not serious	not serious	serious ^g^	none	5/63 (7.9%)	17/83 (20.5%)	RR 0.39 (0.15 to 0.99)	**125 fewer per 1000** (from 174 fewer to 2 fewer)	⨁⨁◯◯ Low
**Severe Anemia (Blood Hemoglobin < 9 mg/dL)**											
2	RCTs	serious ^d^	not serious	not serious	serious ^h^	none	6/126 (4.8%)	18/232 (7.8%)	RR 0.88 (0.30 to 2.56)	**9 fewer per 1000** (from 54 fewer to 121 more)	⨁⨁◯◯ Low
**Recovery from SAM**											
3	RCTs	not serious	serious ^i^	not serious	serious ^j^	none	1096/1696 (64.6%)	1386/1985 (69.8%)	RR 0.91 (0.84 to 0.99)	**63 fewer per 1000** (from 112 fewer to 7 fewer)	⨁⨁◯◯ Low
**All-cause mortality**											
3	RCTs	not serious	not serious	not serious	serious ^k^	none	135/1696 (8.0%)	149/1990 (7.5%)	RR 1.30 (0.87 to 1.95)	**22 more per 1000** (from 10 fewer to 71 more)	⨁⨁⨁◯ Moderate
**Withdrawal from the study**											
3	RCTs	not serious	serious ^l^	not serious	serious ^m^	none	371/1696 (21.9%)	381/1985 (19.2%)	RR 1.25 (0.98 to 1.60)	**48 more per 1000** (from 4 fewer to 115 more)	⨁⨁◯◯ Low

Abbreviations: CI: confidence interval; MD: mean difference; RCT: randomized controlled trial; RR: risk ratio, SAM: severe acute malnutrition, SMS: soya–maize–sorghum, WHO: World Health Organization. Interpretation of certainty ratings: very low (we have very little confidence in the effect estimate), low (we have limited confidence in the effect estimate), moderate (we have moderate confidence in the effect estimate; the true effect is likely close to the estimate of the effect), or high (we have high confidence that the true effect lies close to that of the estimate of the effect). ^a^ Both of the included studies in this analysis were at a high risk of bias for this outcome. The data from both studies were available only for a subset of the patients. ^b^ Even though the unexplained statistical heterogeneity was 52%, based on *I*^2^ statistics, the direction of effect was in the same direction. We did not downgrade for inconsistency. ^c^ The overall effect seems to be small. Even though the confidence interval of the summary estimate did not include a null effect, the lower limit of the confidence limit was very near to the null effect. The results of the blood hemoglobin from both studies were not adjusted for ethnicity or altitude. ^d^ Both the included studies were at high risk of bias due to lack of data for the full set of study participants. ^e^ Even though the confidence interval around the summary estimate did not include 1, the upper limit approached a null effect. In addition, the values of hemoglobin were not adjusted for altitude and ethnicity. ^f^ The only included study was at high risk of bias due to data available only for a subset of patients included in the study. ^g^ The analysis included only one study with a total of 22 events in both groups. The confidence interval around the summary estimate was wide. In addition, the upper limit of the confidence interval of the summary estimate reached almost a null effect. ^h^ The confidence interval of the summary estimate was wide and included a null effect. ^I^ Even though the direction of effect was in favor of a high iron dose in RUTF, the magnitude of effect differed among the included studies. The unexplained statistical heterogeneity was 76%, based on *I*^2^ statistics. ^j^ The upper limit of the confidence interval around the summary estimate almost reached a null effect. ^k^ The confidence interval around the summary estimate included a null effect with the possibility of a beneficial effect or an increased risk of mortality. ^l^ The magnitude of the effect varied among the studies. The *I*^2^ was 60%. ^m^ The confidence interval around the summary estimate included a null effect with the possibility of a beneficial effect or an increased risk of withdrawal from study.

**Table 5 nutrients-14-03116-t005:** Description of side-effect profiles in the included studies.

Author	Side-Effect Criterion	Unit	Value in High-Iron RUTF Intervention Group	Value in WHO Standard-Iron RUTF Comparison Group	*p*-Value	Notes
Irena 2015 [25]	Percentage of children who reported at least one episode of diarrhea	% (n)	20.0 (9)	15.6 (7)	0.6	All of the children on P-RUTF who reported skin rash were from the same health center, and the rash was not specific to certain body parts; it was itchy and appeared as mild with no pustules or vesicles.
	Percentage of children who reported vomiting	% (n)	4.4 (2)	6.7 (3)	1.0	
	Percentage occurrence of skin rash	% (n)	–	13.3 (6)	–	
Bahwere 2016 [12]	Percentage of children with side-effects related to RUTF intake	% (n/N)	2.74 (2/73)	2.22 (2/45)	0.862	No serious side-effects were detected, and no reasons for interrupting the study were identified. No difference was noted in rates of diarrhea, fever, or abdominal pain, and data were the same for children <24 months and >24 months.
Akomo 2019 [4]	Percentage of children with inflammation—adjusted plasma ferritin at discharge > 100 μg/L, indicative of excess iron reserve	% (n/N)	1.6 (1/64)	4.8 (4/84)	0.559	There was no effect of iron content on risk of iron overload or gut inflammation. Complaints of fever, diarrhea, or cough were rare in all study arms in both age groups, with a comparison of median values showing no statistical differences.

Abbreviations: CI: confidence interval; BIS: body iron stores; FSMS-RUTF: milk-free soya–maize–sorghum-based ready-to-use therapeutic food; PM-RUTF: peanut-paste-based ready-to-use therapeutic food; RUTF: ready-to-use therapeutic food; SAM: severe acute malnutrition.

## Data Availability

Further details of data analysis and risk of bias assessments are available on request.

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
