# Peer review of "Increased vs. Standard Dose of Iron in Ready-to-Use Therapeutic Foods for the Treatment of Severe Acute Malnutrition in a Community Setting: A Systematic Review and Meta-Analysis"

_nutrients, 2022, doi:10.3390/nu14153116_

Round 1
Reviewer 1 Report
The authors have conducted a systematic review and meta-analysis investigating different iron concentrations in RUTF on child health outcomes, which is important outcomes to consider for the ideal micronutrient composition of RUTFs. The authorship team has significant experience in this methodology and have applied Cochrane methodology to conduct this review in a rigorous manner. Overall, the review is well-conducted; the following suggestions are provided:
In-text reference formatting is inconsistent and bibliographic references should be checked and confirmed.
Title
Is use of optimal justified here? Should the title format be structured as ( [Intervention A] versus [intervention B] for [health problem] ) since the review compared standard vs. high iron?
Introduction
Rationale
More should be added to the introduction to make the biological plausibility of the research question clear.
Paragraph 1, last sentence: 1-2 sentences could be added on additional consequences of iron deficiency on health outcomes.
Paragraph 2: Can the rationale of the review be expanded:
Providing standard and alternative RUTF formulation iron concentrations would help provide scale relative to human requirements, safety limits, and other common iron supplementation dosages.
L43: ‘may be more effective’, can you provide more specific details from these trials?
What other factors vary between the standard and alternate formulations, and would these impact iron absorption?
Zinc and copper blood concentrations are secondary outcomes. Can more information be added for these outcomes and their interaction with iron and iron outcomes? I see now that no data was found. Recommend author’s discretion for including rational for prespecifying these outcomes.
Objectives
2.1 Study type: list of study types could be re-written to improve reading flow
2.2 Population: Are there other criteria or cut-offs that studies have used to define SAM, and if so, can the authors clarify why those would not be considered for this review?
Please correct WHO reference 19.
Methods
L75, 84: Confirm units for WHO standard iron. Per 100 g and 100 mg are both listed.
2.6 Literature Search: Although all search strategies are indicated as published in the protocol, please include the primary search strategy in this manuscript.
Were all databases searched on June 24, 2021? – Please clarify in line with PRISMA.
Results
Table 1 requires formatting
Can study characteristics (3.3 – 3.5) just be presented in Table 1 and any key aspects summarized in the text? Currently there are significant redundancies.
Table 3 – please specify units of ratios (e.g., mass/mass or mol/mol)
Table 4 is difficult to read.
3.8.1 – For the risk of bias due to only reporting on a subset, please add what % of the study population was in the subset for each study.
Adjusting for altitude/ethnicity: is the approximate altitude reported in each study? This could help interpret how significant lack of adjustment would bias results.
How would using altitude adjusted Hb from the Akomo 2019 impact the meta-analysis?
3.9.3 Since meta-analysis was conducted on weight gain, should weight gain also have GRADE reported even thought it wasn’t included in the outcomes?
Total MD for weight gain in text and fig. 9 do not seem to match. (Figure includes 0 in confidence interval but text does not).
Discussion
4.4 The sentence “We wrote a protocol before the start of the review that was externally re-545 viewed and publicly available before the study18.” should reflect the date of protocol submission (14 Sept, 2021) was after the search (June 24, 2021).
Please check references, esp 16, 19
Check table of contents and pg. numbers in supplemental material.
Reviewer 2 Report
This is an excellent study, the topic is of global importance and the manuscript is well written. The detailed Methodology is based on accepted practice for systematic reviews/meta analyses and helps provide a good framework for better understanding the study and also a template for future research. The Results are clearly reported.
Several recommendations for the Discussion are:
1) Include further discussion on how this study compares to other studies in the literature. Currently the Discussion section is primarily a review of the study results, critique of the completeness/certainty of evidence, and potential bias of the review process.
2) In the Implications for Practice, add further information/recommendations on how practitioners could apply the study results to future RUTF formula development and to delivery/use of RUTFs with targeted populations.
